# Rational Food Design Targeting Micronutrient Deficiencies in Adolescents: Nutritional, Acoustic-Mechanical and Sensory Properties of Chickpea-Rice Biscuits

**DOI:** 10.3390/foods12050952

**Published:** 2023-02-23

**Authors:** Clara Talens, Laura Garcia-Fontanals, Paula Fabregat, Mónica Ibargüen

**Affiliations:** 1AZTI, Food Research, Basque Research and Technology Alliance (BRTA), Parque Tecnológico de Bizkaia, Astondo Bidea, Edificio 609, 48160 Derio-Bizkaia, Spain; 2Basque Culinary Center, Facultad de Ciencias Gastronómicas, Mondragon University, 20009 Donostia–San Sebastián, Spain

**Keywords:** micronutrient deficiencies, rational food design, adolescents, biscuits, texture, chickpea flour, targeted nutrition

## Abstract

“Hidden hunger”, the deficiency of important mineral micronutrients, affects more than 2 billion people globally. Adolescence is unquestionably a period of nutritional risk, given the high nutritional requirements for growth and development, erratic or capricious diets and the increased consumption of snacks. This study applied the rational food design approach to obtain micronutrient-dense biscuits by combining chickpea and rice flours to achieve an optimal nutritional profile, crunchy texture and appealing flavour. The perception of 33 adolescents regarding the suitability of such biscuits as a mid-morning snack was examined. Four biscuits were formulated, with different ratios of chickpea and rice flours (CF:RF): G100:0, G75:25, G50:50 and G25:75. Nutritional content, baking loss, acoustic-texture and sensory analyses were carried out. On average, the mineral content of biscuits with the CF:RF ratio of 100:0 doubled compared with the 25:75 formula. The dietary reference values for iron, potassium and zinc reached 100% in the biscuits with CF:RF ratios of 50:50, 75:25 and 100:0, respectively. The analysis of mechanical properties revealed that samples G100:0 and G75:25 were harder than the others. Sample G100:0 showed the highest sound pressure level (Smax). Sensory analysis showed that increasing the proportion of CF in the formulation augments the grittiness, hardness, chewiness and crunchiness. Most of the adolescents (72.7%) were habitual snack consumers; 52% awarded scores ≥ 6 (out of 9) to biscuit G50:50 for its overall quality, 24% described its flavour as “biscuit” and 12% as “nutty”. However, 55% of the participants could not pinpoint any dominant flavour. In conclusion, it is possible to design nutrient-dense snacks that meet the micronutrient requirements and sensory expectations of adolescents by combining flours naturally rich in micronutrients.

## 1. Introduction

The new WHO European Regional Obesity Report 2022, published on 3 May 2022 by the WHO Regional Office for Europe, reveals that overweight and obesity rates have reached epidemic proportions across the region and are still escalating. None of the 53 member states is on track to meet the WHO Global Noncommunicable Disease (NCD) target of halting the rise of obesity by 2025 [1].

Early studies from several countries in the region indicate that the prevalence of overweight and obesity and high mean body mass index have increased in children and adolescents during the COVID-19 pandemic [2,3]. Moreover, adolescence is unquestionably a period of nutritional risk given the increased nutritional requirements for growth and development, erratic or capricious diets and the increased consumption of snacks, fast food and refreshing beverages. During this period, deficiencies of specific minerals such as Ca, Fe and Zn and vitamins such as A, D, B6, B12, riboflavin, niacin, thiamine and folic acid are common. Low intake of fibre and complex carbohydrates has also been noted [4].

The AVENA study (*Alimentación y Valoración del Estado Nutricional en Adolescentes*) conducted in 2012 with Spanish adolescents revealed that certain healthy dietary habits (i.e., mid-morning snack, afternoon snack, more than 4 meals per day, adequate eating speed) were associated with low body fat [5]). Moreover, public health recommendations in paediatric scientific journals suggest that most children should eat between four and six times a day [6,7].

Snacks can account for up to a third of the daily energy intake. Thus, it is of great interest to the food industry to provide snacks of high nutritional quality. This could be achieved by adding protective factors and avoiding risk factors (for infants and adolescents), so the products might be recommended as meeting nutritional requirements [8]. To improve the nutritional quality of snacks, one should reduce risk factors such as sugar, salt, refined grain flour and energy content. Likewise, the proportion of nutritionally valuable ingredients providing protein, fibre and micronutrients should be increased.

Legumes are a staple food in many Asian, African and Mediterranean countries. “Hidden hunger”, the deficiency of important mineral micronutrients, affects more than 2 billion people globally [9]. Naturally rich in micronutrients (iron and zinc), legume flours could be used to formulate recipes for nutritionally rich and balanced biscuits. The cereal industry often uses rice as a substitute for wheat. Refined rice flour has nutritional disadvantages as it is high in carbohydrates but low in protein, fibre and micronutrients. However, rice flour could be combined with other ingredients, such as chickpea flour, to develop value-added snacks with a rich and well-balanced nutrient composition [10,11].

Chickpeas are highly nutritious due to their low lipid content (2.7–6.48%), rich in polyunsaturated fatty acids (66%), and high content of protein (17–22%), starch (52.5%), dietary fibre (18–22%), bioactive compounds and essential vitamins and minerals. These legumes are a good source of calcium, potassium, magnesium, iron, zinc and important vitamins such as riboflavin (B2), niacin (B3), thiamine (B1), folic acid (B9), β-carotene (vitamin A precursor), vitamin E, vitamin C, pantothenic acid (B5) and pyridoxine (B6). Therefore, chickpea products can complement the vitamin pool supplied by other foods [12]. The demand for chickpeas as a functional ingredient in food production is on the increase. Chickpea flour has been incorporated into a variety of goods, such as breads, biscuits, pasta, snacks and dairy, to improve their nutritional value [11]. Goñi and Valentín-Gamazo [13] have shown that adding 25% of chickpea flour to wheat pasta decreases the glycaemic index and increases the mineral, fat and resistant starch content. Moreover, the consumers of chickpeas and/or hummus have a higher nutrient intake of dietary fibre, polyunsaturated fatty acids (PUFAs), vitamins A, E and C, folic acid, magnesium, potassium and iron than non-consumers [14]. Importantly, chickpeas have high levels of the micronutrients usually lacking in diets consumed by adolescents.

The term Rational Food Design (RFD) has been used in previous research to refer to the design of food products with specific functionalities to satisfy “the needs and desires” of the consumer [15]. The official meaning of the word “rational” is “based on facts and reason”, therefore, RFD should rely on any type of scientific and technological knowledge to design food formulas and food structures. Previous studies have applied RFD principles to create high-quality healthy foods by including in the RFD approach imaging technologies, such as atomic force microscopy [16], or microtechnology for testing ingredient functionality [17]. However, to the best of our knowledge, the inclusion of the dietary reference values (DRV), provided by EFSA, as formula design variables, and acoustic-texture measurements, as response variables, during the RFD process has not been approached yet.

The aim of this study is twofold. Firstly, the study will include EFSA’s DRV for adolescents from 11 to 14 years, engaged in all types of physical activity, using the RFD approach to obtain micronutrient-dense biscuits by combining chickpea and rice flours, to achieve an optimal nutritional profile, crunchy texture and appealing flavour. Secondly, the perception of adolescents regarding the suitability of such biscuits as a mid-morning snack will be examined.

## 2. Materials and Methods

### 2.1. Rational Food Design

In this study, we suggest a widening of the use of RFD for targeted nutrition (aimed at the nutritional needs of specific groups). For this purpose, RFD could be applied in two steps of the food development process:1.during the formula design step, which is also part of the process of structure design, when two important decisions are made: (1) the target nutritional composition of the new or improved formula; (2) the ingredients and raw materials used based on their macro- and micronutrients composition.2.during the step preceding the consumer tasting, when a selection of the optimum samples has to be made based on acoustic-textural and sensory analysis.

The rationale applied during the formula design was based on EFSA’s DRV. For adequate energy distribution, the daily food intake should be supplied in 5 meals (Table 1). The proportions should be as follows: 20% at breakfast, 10–15% at mid-morning meal, 25–30% at lunch, 10–15% at evening snack and 25% at dinner. Eating between meals should be avoided [6,7,18].

Many scientific groups and societies recommend distributing energy and nutrients among four to five daily meals to improve health [19,20]. According to the American Heart Association [21], the mid-morning snack should be easily digestible and not excessive in calories to sustain the feeling of satiety and reach lunchtime with a sufficient but not unmanageable appetite. Table 2 shows the nutritional requirements for mid-morning snacks based on the European Food Safety Authority (EFSA) dietary reference values (DRV) for the EU [22]. The target population are healthy individuals of both genders, aged 11–14, engaged in all types of physical activity.

Therefore, although mid-morning snacks can reduce appetite during lunch, it is crucial to consider their nutritional quality.

### 2.2. Ingredients

Chickpea flour (Don Pedro) (CF) was obtained from Legumbres Pedro S.L. (Cadiz, Spain). The flour contained 15.0 g of moisture, 53.5 g of carbohydrates, 4.4 g of sugar, 20.5 g of protein and 6.6 g of fat (per 100 g).

Rice flour Remyflo R 200 T (RF) was purchased from BENEO-Remy N.V. (Leuven, Belgium). The composition per 100 g was 10.0 g of moisture, 81.0 g of carbohydrates, 8 g of proteins and 1 g of fat.

Sunflower oil, orange blossom water, sugar and salt were bought at the local supermarket (Makro, Derio, Spain).

### 2.3. Biscuit-Making Procedure

Four biscuits were formulated using different ratios of chickpea flour and rice flour (CF:RF). These ratios were: 100:0, 75:25, 50:50, 25:75. The biscuits were made following the formulations, in which only chickpea and rice flour percentages changed. The rest of the ingredients were kept constant (Table 3).

All ingredients were weighed using high-precision (±0.0001 g) scales AB304-S (Mettler Toledo, Greifensee, Switzerland). The powdered ingredients were mixed using a planetary mixer Sammic BM-5 (Sammic S.L., Azkoitia, Spain), for 30 s at 202 rpm. Then, water was added and the mixture was stirred for 10 s at 260 rpm. Finally, oil was mixed in for 30 s at 202 rpm. Batches of 500 g were produced in triplicate. The dough was removed from the mixer and allowed to rest for 10 min. It was processed further using a sheeting machine Sammic FMI-31 (Sammic S.L., Azkoitia, Spain). The dough sheets were flattened manually with a wooden rolling pin to an approximately 2.5-mm thickness and then cut into circles of 4.5 cm in diameter. The biscuits were baked in an electric oven (MIWE Condo type CO 2 0608, MIWE GmbH, Arnstein, Germany) at 150 °C for 25–35 min and then at 120 °C for 20–30 min. The biscuits were cooled for 30 min on a rack and stored in airtight containers until evaluation.

### 2.4. Physico-Chemical Analysis

Powdered samples of the four biscuit types were analysed according to the ISO standards for moisture (ISO 1442:1997), protein content (ISO 937:1978), crude fat (ISO 1443:1973), fatty acids profile (by chromatography), salt content estimated by analysis of chloride by potentiometric titration with silver nitrate solution, and total sugars (ISO 22184:2021). Total dietary fibre was determined by the AOAC enzymatic gravimetric method 991.43. The carbohydrate content was determined by difference. After proximate analysis, energy values were calculated using the Atwater general factors (4 kcal/g for protein and carbohydrates, 9 kcal/g for fat and 2 kcal/g for fibre). Samples were analysed in triplicate. The mineral composition (calcium, phosphorus, iron, magnesium, manganese, zinc, potassium and copper) was examined following the standard DIN EN 16943:2017-1. The selenium content was obtained following DIN EN 15763:2010-04. Sodium levels were established using gas chromatography–mass spectrometry (LC-MS/MS).

The Water activity was measured using a water activity meter (AquaLab PRE, Lab Ferrer, Cervera, Spain).

The baking loss (%) was determined on three independent samples using Equation (1), 24 h after baking, where W_f_ is the weight of the sample after baking and W_0_ is the weight before baking:% Baking loss = (W_f_ − W_0_)/W_0_ × 100 (1)

### 2.5. Instrumental Texture Analysis

The mechanical and acoustic properties of the biscuits were simultaneously measured employing a Texture Analyser (TA.HDplus, Stable Micro System Ltd., Surrey, UK) and a microphone in combination with a Deltatron preamplifier (Brüel Kjær, Nærum, Denmark). A 30-kg load cell was used and the key parameters were extracted employing the Exponent software (v.6.1.16.0, Stable Micro System Ltd.). Each biscuit was placed upside down on a perforated surface. The samples were penetrated using a cylinder probe of 4 mm in diameter at a test speed of 1 mm/s. The distance was 10 mm, and the trigger force was 5 g, based on the procedure of da Quinta, Alvarez-Sabatel [23]. Ten replicates were used for each biscuit type.

During the test, the acoustic emission was registered by a microphone calibrated using a calibrator type 4231 (94- and 114-dB sound pressure level [SPL], 1000 Hz) (Brüel Kjær). The distance of the microphone to the sample was 10 mm with an angle of 45° [24]. The filter function of the preamplifier screened out the background noise. No simultaneous activities were carried out in the laboratory to avoid noise interference.

The mechanical properties of the samples were determined using the following parameters: the maximum force required to break (N) as a direct measure of the hardness and the probe distance needed to break the sample (mm) as an indicator of the fracturability [25]. The crispness attribute was associated with several acoustic and mechanical parameters [23,26,27]. These were the maximum loudness perceived during the break evaluated as sound pressure level (SPL) in dB, number of sound peaks (peaks higher than 1 dB), number of force peaks (peaks higher than 0.2 N) and the linear distance (N*s). This last parameter, used as an indicator of jaggedness, was calculated as the length of an imaginary line joining all force peaks in a force–time graph [28].

### 2.6. Sensory Evaluation by Trained Assessors

A panel of seven assessors trained in quantitative descriptive analysis (QDA) of biscuits evaluated the samples. The panel was selected and trained following the ISO 8586:2012 procedure. Six 1-h sessions were performed for descriptor development, definition and training. Training sessions utilised reference standards. The attribute values were recorded using a 5-point scale. Finally, the four samples were evaluated in duplicate. The samples were labelled with a random three-digit numeric code and presented monadically in a randomised and balanced order. Still water was served for palate cleansing between samples.

The sensory evaluations were carried out at the Sensory Science Laboratory in AZTI (Derio, Spain) in individual booths designed in accordance with ISO 8589:2007.

### 2.7. Consumer Tasting

The sample consumer population was selected from visiting students from a local secondary school (11–12 years of age). An informed consent form was sent to the parents before the tasting session, indicating the objective of the study and listing the allergens. Thirty-three students participated; 33% were female, and 67% were male. The objective of the study was (i) to carry out a sensory evaluation of the newly developed biscuit and (ii) to establish whether they could identify chickpea flavour. The group assessed only one sample. The selection of the sample was based on the results of the texture measurements and trained panel sensory analysis.

The participants assessed the samples in individual cabins illuminated by fluorescent lamps. The sample was served in individual plates. Participants were asked to record their liking intensity scores for overall appearance, overall colour, overall aroma, overall texture and overall flavour (see Appendix A). A 9-point hedonic scale was used (9 = like extremely and 1 = dislike extremely). The overall liking was also assessed, together with the open question, “Are you able to identify a particular flavour?”. Finally, a questionnaire with two consumption habit questions was presented. The first question was, “Do you consume snacks?”. The second was a multiple-choice question, “How often do you eat biscuits?”. The options were: twice a week or more often, once a week, twice a month, once a month, once every 2–3 months or less frequently.

### 2.8. Statistical Analysis

The data were processed statistically using the software package XLSTAT 2019.1.2 (Addinsoft, Boston, MA, USA). Analysis of variance (ANOVA) and Tukey’s HSD test for comparison of sample means were used to identify nutritional properties and instrumental texture parameters that significantly differed between the samples. All data were expressed as means ± standard deviation (SD). The average sensory configuration obtained by the panel is displayed (as for Principal Component Analysis (PCA)) on a score plot representing the inter-product sensory distances.

Descriptive analysis was performed using the liking data recorded by consumers (the overall liking data and the individual attribute liking data reported on the hedonic 9-point scale).

## 3. Results

Results showed that, by using combinations of chickpea and rice flours, it was possible to apply a rational food design to create nutrient-dense and sensory-appealing biscuit-like structures with an improved nutritional profile (targeting EFSA’s DRV for adolescents 11 to 14 years, engaged in all types of physical activity), compared to the average profile of 49 plain biscuits found in the Mintel database.

### 3.1. Nutritional Composition

The mean values for the nutritional properties of the four different biscuits can be found in Table 4.

The protein content varied from 7.71% to 13.92%, with the sample G100:0 showing the highest value and the G25:75 the lowest. Samples G75:50 and G50:50 did not significantly differ in their protein content (10.54% and 9.89%, respectively). These results indicate that the relative amount of protein in the biscuits increases with increasing chickpea flour (CF) content. This is not surprising, as rice flour (RF) contains less protein than CF.

A similar trend was observed for the fat content, with values ranging between 19.92% and 21.15%; however, the differences between samples were smaller and not significant. These results reflect the lower fat content of RF compared to CF.

The carbohydrate content ranged from 49.63% to 62.27%; the sample G25:75 had the highest level of carbohydrates, which was the lowest in sample G100:0 (the difference was significant). Samples G75:25 and G50:50 had intermediate profiles with 56.15% and 56.56% carbohydrate content, respectively. The carbohydrate content of the biscuits increases with rising RF content, which is consistent with the higher carbohydrate levels of RF compared to CF. However, a tendency for the sugar and fibre content to increase was observed for biscuits with larger amounts of CF. Sample G100:0 contained the most sugar and fibre (18.84% and 7.07%, respectively) and sample G25:75, the smallest amounts (11.60% and 2.59%). These differences were statistically significant. Thus, the sugar and fibre concentrations rise with increasing CF content. These observations are consistent with the differences in nutritional characteristics of these two flours [29].

No significant differences were seen between the salt contents of the different formulations, indicating that changes in the flour ratio did not affect this nutrient.

The data on the mineral content of the biscuits showed that the concentrations (in mg/100 g) of iron (1.26–3.39), potassium (233.74–576.70), zinc (1.00–1.99), phosphorus (111.67–209.73), magnesium (54.12–113.45), copper (0.25–0.62), manganese (0.93–1.10) and calcium (14.23–29.00) rise significantly with increasing CF content. On average, the mineral content increased by 2-fold when increasing the CF:RF ratio from 25:75 to 100:0.

For sodium, no significant differences were observed. For selenium, the concentration increases significantly with rising RF content, varying from 5.83 µg/100 g for the sample G100:0 to 9.08 µg/100 g for the G25:75. These results are consistent with the mineral content of chickpea and rice flours [29].

The energy content did not differ significantly between the samples. The energy values per 100 g ranged from 471.76 kcal for the sample G100:0 to 457.15 kcal for the G25:75. There is a tendency for the energy content to increase as the proportion of added CF increases.

### 3.2. Instrumental Texture

The results of the instrumental texture analysis, as well as the baking loss, moisture content and water activity of the four different biscuits can be found in Table 5.

Samples G100:0, G75:25 and G25:75 did not show significant differences between their percentages of baking loss (29.26%, 29.21% and 28.89%, respectively). The sample G50:50 showed greater baking loss (30.07%) than G25:75. The moisture content of samples G100:0, G75:25 and G50:50 did not differ significantly (3.96%, 3.59% and 3.95%, respectively); the G25:75 showed increased moisture content (6.50%). The highest water activity (a_w_) was observed for the sample G25:75 (0.47), followed by G100:0 (0.31), G50:50 (0.29) and G75:25 (0.27). Therefore, when the relative RF content was greater than 50%, the moisture content and a_w_ of the biscuits increased. Changing the flour ratio did not affect the baking loss.

The analysis of mechanical properties revealed greater maximum force (Fmax) for the samples G100:0 and G75:25 (18.13 N and 19.68 N) than for G50:50 and G25:75 (11.68 N and 10.97 N, respectively). When the mixture contained more CF than RF, the hardness of the biscuit increased. However, all the samples showed similar distances at break (ranging from 0.83 mm to 0.70 mm) with no significant differences; they all had similar fracturability (or fragility) despite the differences in hardness. Sample G100:0 showed a significantly greater number of force peaks (NFP) (9.80) than samples G75:25, G50:50 and G25:75 (1.17, 2.60 and 2.20, respectively; differences not significant). Accordingly, sample G100:0 probably suffered more breaking events. Moreover, the sample G100:0 showed greater linear force peak distance (LDF) (94.76 N*s) than the other samples and sample G75:25 had significantly greater linear distance (81.35 N*s) than G50:50 (68.48 N*s). This could indicate that, as the CF content increased, the LDF also tended to increase. The sample G25:75 showed an LDF of 76.40 N*s, not significantly different from G75:25 and G50:50, like the NFP. This result might reflect the relationship between the NFP and LDF; the more fluctuations in force, the longer the line joining the force points.

The highest value of maximum sound pressure level (Smax) was obtained for sample G100:0 (75.74 dB), significantly higher than for samples G75:25 and G25:75 (65.36 dB and 67.99 dB). In contrast, no significant differences were detected between the numbers of sound peaks (NSP) for the four different biscuits.

Since the parameters LDF, NFP, NSP and Smax are the indicators of the crispness of a product [23,26,27], these results suggest that, when the biscuit is made with 100% chickpea flour, its crispness will be significantly higher than when the mixture of the two flours is used.

Figure 1 shows force-time and sound-time curves obtained in the texture and acoustic event analysis for each biscuit. The graph for G100:0 (as an example) shows a force increase region, starting from the first contact between the probe and the biscuit until the first major drop in the force (at around 1 s). Within this region, the compression force increased almost linearly with the displacement, while acoustically it was very quiet. This suggests that the biscuit undergoes deformation but no major structural damage. The acoustic signals recorded in this linear domain were not considered, as they resulted from surface contact between the biscuit and the probe. Then, the compression force became jagged and many acoustic events were recorded within a very short period. This is when the structural breakdown occurs, at the first crack. There was no further increase in the compression force after that break, but rises and falls in the measured force could be observed. These force reductions reflect the ongoing minor structural fracture in the biscuit. At around 2 s, a sharp drop in the compression force occurred, which corresponded to the major structural breakdown. After this significant event, the biscuit remained on the test platform and it took another push for the fractured biscuit to finally fall to the texturometer base. At this point, the compression force reached zero. The acoustic signals recorded after this major breakdown were not considered [24].

We focused on acoustic signals in the jagged region of the force–time curve (between 1 and 2 s) (Figure 1, sample G100:0), where the biscuit breaking occurred. A group of acoustic events can be observed for each major force drop. Since these events did not gradually decrease in intensity and had no periodic pattern, they were probably not due to sound echoes or resonances. It is most likely that they reflected a series of structural element fracture events captured within a major force peak. It can be hypothesised that the energy dissipated from the biscuit break will spread out, probably in the form of sound. Therefore, the Smax should correspond to the energy released by the major structural breakdown [24].

Figure 1 shows that, for each drop in the compression force (force peak), an acoustic signal (sound peak) was detected, demonstrating the links between the acoustic events and the decrease in the force of a single break event (corresponding to the dissipated energy from the break). A correlation between the major structural breakdown (Fmax) and the Smax can also be observed.

### 3.3. Sensory Evaluation and Consumer Liking

#### Sensory Profile of Biscuits

The panel generated seven descriptors to describe the biscuits; five referred to the texture (hardness, grittiness, fragility, prickliness and chewiness), and two to acoustic sensations (crispness and crunchiness) (Table 6).

Figure 2 presents the results of the PCA of the data generated by the sensory panel for the four biscuit formulations. Axis F1 explained 49.09% of the sensory variation between the biscuits and Axis F2, 42.16%. The results indicated that the attributes “grittiness”, “fragility”, “hardness”, “chewiness” and “crunchiness” had discriminative power (ordered from the largest to the smallest, *p*-values < 0.01). However, the “crispiness” and “prickliness” attributes could not be used to distinguish the samples from each other because they did not have discriminatory power (*p*-values of 0.13, 0.43, respectively) and, consequently, did not appear in the PCA analysis (Figure 2). Figure 2 shows the vector (red line) corresponding to each attribute and the four biscuit samples (blue points).

In the PCA graph, the samples close to each other have similar sensory profiles, and larger distances indicate increased sensory differences. The evaluated sensory attributes are represented by vectors. The vector resultants help to characterise the samples: the higher the resultant on an axis, the higher the discriminating power of the attribute.

Table 7 presents the results of assessing the samples included in the PCA. The sample G100:0 differed from all the others; it showed the lowest fragility and the highest crunchiness and chewiness values. The G50:50 was the least crunchy and chewy and the most fragile. The G100:00 and G50:50 samples showed similar degrees of grittiness and hardness, greater than G75:25 and G25:75. In contrast, the sample G25:75 stood out from the others by showing the lowest grittiness and hardness. The sample G75:25 showed an intermediate sensory profile for grittiness, hardness, crunchiness, chewiness, and fragility, even though it had the crispiest texture. The fragility, crunchiness, and chewiness of G25:75 and G75:25 biscuits were very similar (no significant differences). Their fragility was lower than for G50:50 and greater than for G100:0 samples. The crunchiness and chewiness of G25:75 and G75:25 were lower than for the G100:00 sample and greater than for G50:50.

The samples G100:00 and G25:75 were very far from each other in the graph (Figure 2), indicating large sensory differences. Moreover, the separation between these samples occurs on Axis F2. This means that the differences between them were explained by the attributes whose resultant vector was located on this axis (grittiness, hardness and chewiness). Therefore, it can be concluded that changing the amount of CF in the formulation directly affects the texture of the biscuit, particularly its grittiness, hardness and chewiness. The grittiness, hardness and chewiness of the biscuit increase along with the growing amount of CF. Samples G75:25 and G50:50 were closer with respect to Axis F2, so their grittiness, hardness and chewiness were similar. These results were consistent with the fact that the G75:25 and G50:50 biscuits only differed by 25% of the CF, while G100:0 had 75% more CF than G25:75.

As the samples with larger amounts of CF (G100:0 and G75:25) tended to be less fragile than the others (G50:50 and G25:75), we can conclude that reducing the amount of CF increases the fragility of the biscuits. A greater crunchiness was observed as the ratio CF:RF increases from 50:50 to 100:0.

### 3.4. Consumer Tasting

Only one sample was assessed by the adolescents participating in the study, to avoid peer pressure when comparing samples or preferences. The selection of the sample was based on the results of the acoustic-texture measurements and the sensory analysis with the trained panel. The G50:50 had the best texture for children aged between 10 and 12. It was the easiest to chew (the least effort required to chew the biscuit before swallowing) and the most fragile (the least force needed to break it into pieces) (see Table 5 and Table 7). Considering the results of texture and sensory analysis, the biscuits with a higher proportion of CF (G100:0 and G75:25) could be too hard and difficult to chew and the biscuit G25:75 contained too much moisture.

Figure 3 shows the percentages for liking intensity scores (9-point hedonic scale) for overall appearance, colour, aroma, texture and flavour. The sensory attributes of overall appearance, overall colour, overall aroma and overall texture of the chickpea biscuit were evaluated positively (score ≥ 6) by 97%, 91%, 76% and 52% of the consumers, respectively. Only the overall flavour attribute was evaluated negatively (score ≤ 4) by 55% of the consumers.

Regarding the overall liking results, 52% of the adolescents gave a positive sensory evaluation of the overall quality of the chickpea biscuit (sum of choice percentages with scores ≥ 6). However, 33% of participants gave this a negative assessment (sum of choice percentages with scores ≤ 4), stating that the texture seemed a bit too hard and that the biscuit did not have much flavour.

These results indicate that the overall liking of the biscuit was negatively affected by unappealing flavour and texture.

Adolescents participating in this study were not able to identify the main flavour of the biscuits. Some (24%) described the chickpea biscuit flavour as “biscuit flavour”, and 12% of the participants perceived a “nutty flavour”. However, 55% were unable to pinpoint any dominant flavour, answering “no”, “I do not know” or “I cannot”, and 9% described other flavours.

The answers to the two consumption habit questions (the frequency of snack consumption) revealed that 72.7% of the participants consume snacks, most frequently in the form of biscuit and breadsticks. Biscuits were eaten by 14.3% of the consumers several times a week, 9.5% once a week and 14.3% once a fortnight. The remaining 61.8% indicated that they ate biscuits once a month or less frequently.

## 4. Discussion

Table 8 shows the average nutritional content of 49 different plain biscuits, obtained by searching the Mintel Database (October 2022).

Based on the average serving size from the Mintel search (Table 8), a standard 30 g serving size was assumed for the biscuits as mid-morning snack. It was possible to reach the percentages of the reference daily intake (indicated as AR or AI) of micronutrients recommended by EFSA, for adolescents from 11 to 14 years, engaged in all types of physical activity [22], indicated in Table 9.

For none of the four biscuits did the mineral content contained in one serving exceed the Tolerable Upper Intake Level (UL). For all the biscuits, a part of the reference daily intake of the analyzed minerals is covered with the consumption of one serving of biscuits (30 g). Taking into account that mid-morning meal should represent 15% of the daily food intake, and assuming that mid-morning meal should cover approximately 15% of the reference daily intake of minerals, the biscuits should be accompanied by another food that provides minerals to complete this 15%.

Among the top 20 ingredients found in the search (Figure 4), 93% of the biscuits contained flavourings and salt and 86%, raising agents and emulsifiers. Wheat flour and white sugar were present in 71% of the samples. Between 43% and 57% of the biscuits were supplemented with vitamin B6, riboflavin, vitamin B1, niacin, iron, vitamin A, vitamin D or other vitamins and minerals.

Compared to the commercial plain biscuits found in the Mintel search (Table 8, Figure 4), all the chickpea biscuit samples contained less carbohydrates, sugar and salt. However, they had a higher protein and fat (mainly unsaturated) content. The fibre content was augmented in all samples except for G25:75, which contained the lowest amount of CF (the main fibre source). Moreover, chickpea biscuits were rich in potassium, calcium, phosphorus, magnesium, iron, zinc, copper and manganese, and these minerals were naturally present in the ingredients (unlike the additives used in commercial biscuits). As can be seen in Figure 4, most of the vitamins and minerals in plain biscuits targeted at children and sold in Spain between 2017 and 2022 are added to the formulation rather than naturally present in the main ingredients.

The moisture content of the biscuits was similar to the moisture levels reported in other studies of protein-enriched biscuits with CF [30,31]. The only exception was the sample G25:75, in which the industry standard for moisture content in biscuits (1–5%) was exceeded. This high moisture content could be due to the large proportion of starch (approximately 80%) in RF, which might have increased water retention.

Rababah and Al-Mahasneh [30] replaced some wheat flour in biscuits using CF at 3%, 6%, 9% and 12%. For the biscuits enriched with 12% CF, they reported an increase in protein content (from 16.82% for the control to 19.64%) and fat content (from 14.13% to 15.31%). This was to be expected, as the CF contains more protein and fat than the wheat flour. These data are consistent with the current study, where the protein and fat content of the chickpea biscuit increased proportionally to the amounts of CF added. As the chickpea-to-wheat flour ratios increased, the values for most of the liking attributes (overall impression, overall flavour and overall colour) decreased and the hardness of the biscuits increased. As a result, the fortification ratio of 3% gave the best sensory results in descriptive analysis.

In contrast, Yadav and Yadav [31] reported a decrease in biscuit fat content with an increasing degree of wheat substitution with CF and plantain flour. However, this result seemed to be due to the low oil-holding capacity of these flours compared to wheat flour. Moreover, the authors reported an increase in protein content from 7.1% for the 100%-wheat biscuit to 9.2% for the 40%-chickpea biscuit (probably caused by the higher protein content of CF). The team also reported an increase in the amount of fibre in chickpea-enriched biscuits (again, most likely due to the high levels of fibre in CF). An increase in fracture strength, and therefore in hardness, of biscuits with the addition of plantain and CF was also observed (the highest at 40% substitution). This is in agreement with the results of the present study.

Mancebo and Rodriguez [32] added pea protein (up to 20%) to a rice flour biscuit. They found that incorporating this protein decreases the biscuit hardness compared to 100%-rice flour biscuits. Dapčević et al. [33] replaced 10–30% of RF with buckwheat flour, which contains twice the protein of RF and less starch. They found that increasing the relative amounts of buckwheat decreased biscuit hardness and fracturability. Similarly, Gerzhova and Mondor [34] have demonstrated that adding canola protein to 80% rice and 20% buckwheat flour biscuits decreased hardness and increased thickness of the biscuit. Sarabhai and Prabhasankar [35] have reported that adding soy and whey protein to a rice flour biscuit reduced its breaking strength (and, therefore, hardness). The current report does not wholly concur with these studies. Our texture and sensorial results showed that the biscuit hardness, crunchiness and chewiness increased with the rising proportion of CF (and, therefore, protein) added to the formula. This inconsistency may be due to the much higher CF percentages used here in comparison with the studies mentioned above. In those studies, the maximum RF replacement was 30% and the maximum protein concentrate addition was 20%.

The Fmax value was associated with the sensory attributes of hardness, grittiness, chewiness and fragility. Samples G100:0 and G75:25 had higher Fmax values and were considered harder, grittier and more chewy and less fragile than the sample G25:75. This is in agreement with the results of Segnini and Dejmek [36], in whose study the fracture force for a potato chip seemed to be a good predictor of the sensory texture attributes such as hardness, chewiness, crispness (evaluated as crunchiness) and tenderness.

The results of this study suggest that crunchiness was positively associated with the acoustic parameter Smax and the mechanical parameters Fmax, LDF and NFP; as the CF content of the biscuit increased, the values of Smax, Fmax, LDF, NFP and crunchiness tended to rise.

No association between texture and acoustic parameters and the sensory attribute of crispiness was detected. This is in disagreement with the results of da Quinta and Alvarez-Sabatel [23], Gouyo and Mestres [26] and Salvador and Varela [27]. Their studies have reported that the LDF, NFP, NSP and Smax are positively correlated with the crispness of a product. Considering these results, one might expect that the crispness of the 100% chickpea biscuit would be significantly higher than for a product combining the CF with RF. However, the sensory panel did not detect this effect.

Fillion and Kilcast [37] have studied the perception of crispness and crunchiness in fruits and vegetables. They concluded that loudness was not considered when qualifying a product as crunchy or crispy, but it was used to assess the intensity of crunchiness or crispiness. The two attributes involve different frequencies of sound, a low frequency for crunchiness and a high frequency for crispiness. Furthermore, there was no correlation between hardness and crispiness when the hardness was very high. This suggested that a very hard texture could not be perceived as crispy and would be described as crunchy. According to that study, Smax and NSP could not be used to qualify a product as crispy or crunchy since these parameters do not reflect the frequency of the sound but its loudness. Therefore, these acoustic parameters could be related to both crispiness and crunchiness, depending on the product being evaluated.

This might be the reason why the very hard chickpea biscuits were not perceived as crispy (with a minimum score of 0 and a maximum of 2 on a scale from 1 to 5) but rather as crunchy (with a minimum score of 2 and a maximum of 4 on a scale from 1 to 5).

In this work, our approach has been to bring other types of knowledge into the RFD approach for targeted nutrition: the scientific knowledge, provided by EFSA’s DRV, and acoustic-texture measurements to apply RFD targeted to adolescents aged 11–14 years, engaged in all types of physical activity.

Increasing the addition of chickpea flour in a rice biscuit improves its nutritional pro-file, increasing by 2-fold the protein and the mineral content by increasing the CF:RF ratio from 25:75 to 100:0. However, these increase causes a rise in biscuit hardness, grittiness, chewiness and crunchiness, according to texture and sensorial analysis.

## 5. Conclusions

Efforts to develop targeted nutrition strategies for adolescents from 11–14 years should be scrutinized beyond the nutrient contents of food, and include EFSA’s DRV, as well as the acoustic-texture effect, in the case of biscuits. Increasing the addition of chickpea flour in a rice biscuit improves its nutritional profile, increasing by 2-fold the protein and the mineral content by increasing the CF:RF ratio from 25:75 to 100:0. However, these increase causes a rise in biscuit hardness, grittiness, chewiness and crunchiness, according to texture and sensorial analysis. A total of 33 adolescents participated in the study, of whom 52% scored as ≥6 the overall quality of the chickpea biscuit, which was negatively affected by unappealing flavour and texture. This study shows that rational food design is a promising approach to design nutrient-dense and sensory-appealing microstructures aimed at adolescents by combining flours naturally rich in micronutrients. However, the authors acknowledge the limitations of this research in that future work should include the antinutritional factors of the plant-based ingredients used. In addition, including other disciplines, such as imaging technologies, digestion, nutrition and physiological responses, could bring multidimensional perspectives in the design of food matrices that are sensory-appealing and targeted to adolescents.

## Figures and Tables

**Figure 1 foods-12-00952-f001:**
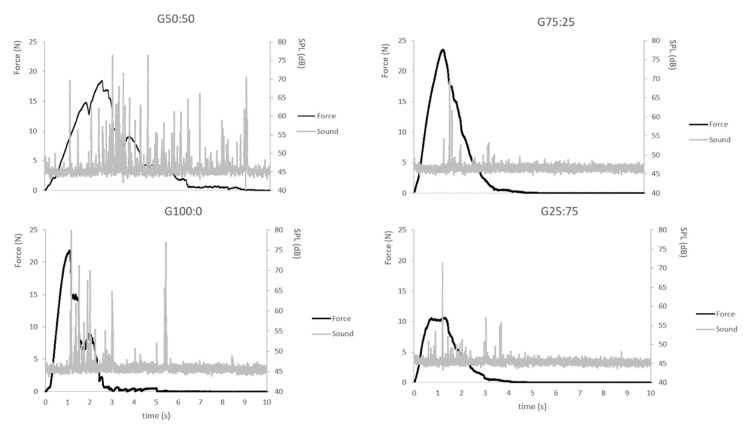
Force–time and sound–time curves for the four chickpea biscuit formulations. Each graph corresponds to a single texture analysis measurement performed for each biscuit. It shows the force and acoustic pressure level (SPL) during the breaking of the biscuit. G100:0 = 100% CF/0% RF; G75:25 = 75% CF/25% RF; G50:50 = 50% CF/50% RF; G25:75 = 25% CF/75% RF.

**Figure 2 foods-12-00952-f002:**
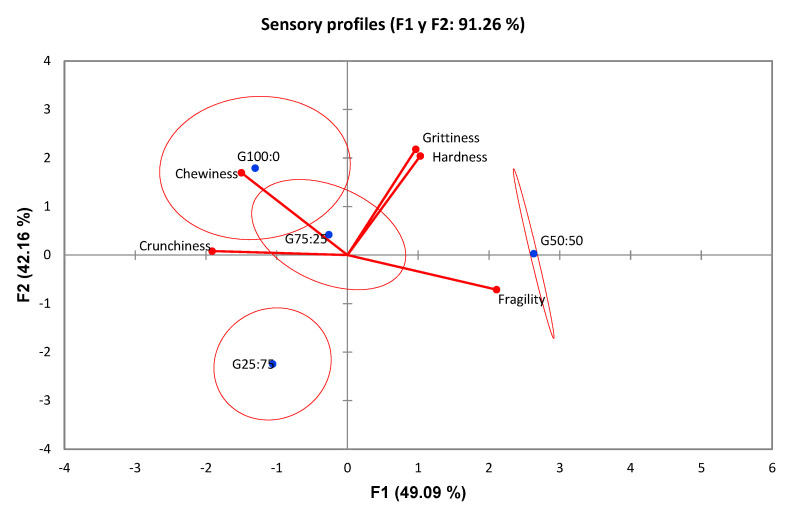
Principal Component Analysis (PCA) of the four chickpea biscuits formulations. G100:0 = 100% CF/0% RF; G75:25 = 75% CF/25% RF; G50:50 = 50% CF/50% RF; G25:75 = 25% CF/75% RF.

**Figure 3 foods-12-00952-f003:**
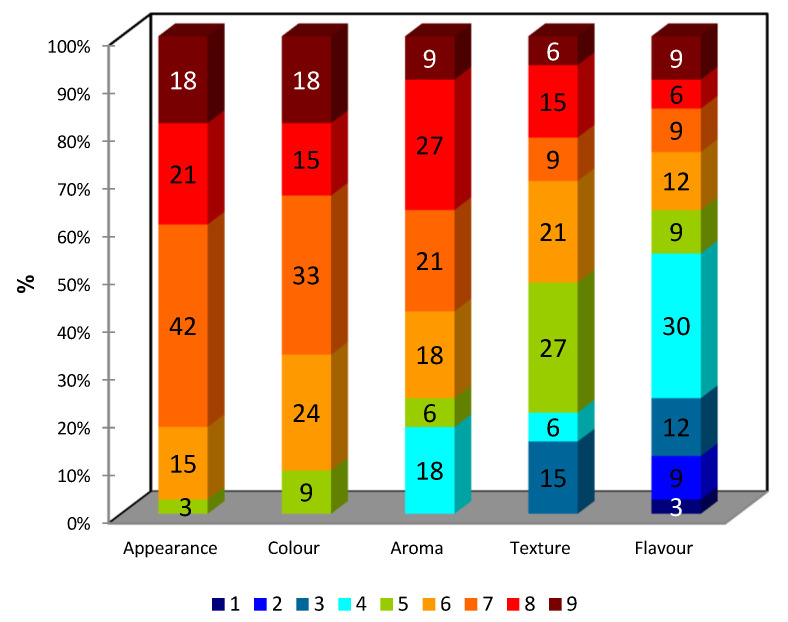
Percentages (values in coloured bars) for liking intensity scores (9-point hedonic scale) for overall appearance, colour, aroma, texture and flavour obtained from consumer tasting.

**Figure 4 foods-12-00952-f004:**
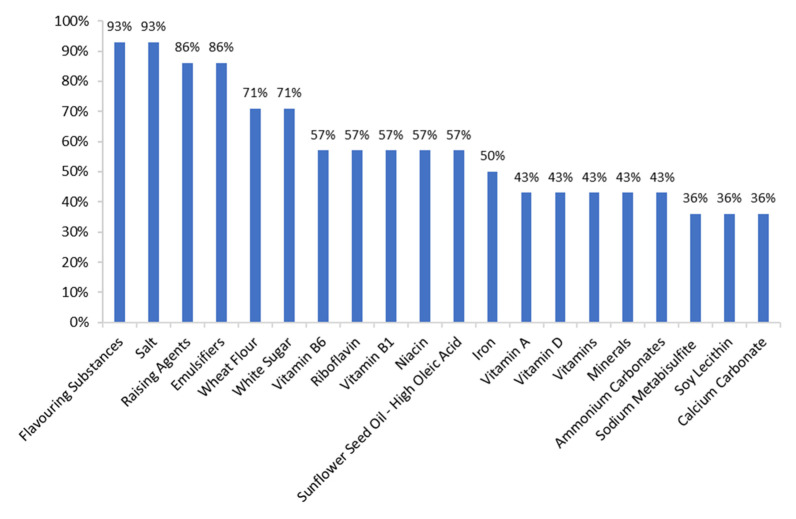
Top 20 (Food) ingredients declared in 49 plain biscuits targeted at children aged 5–13 years, sold in Spain between 2017 and 2022. Data found in Mintel GNPD Database (October 2022).

**Table 1 foods-12-00952-t001:** Recommended energy (% of calories) distribution per meal for adolescents.

Meal	Energy Requirement (Minimum)	Energy Requirement (Maximum)
Breakfast	20%	20%
Mid-morning snack	10%	15%
Lunch	25%	35%
Afternoon snack	10%	15%
Dinner	25%	25%
TOTALTotal	90%	110%

**Table 2 foods-12-00952-t002:** Nutritional requirement for a mid-morning snack (15%) based on EFSA DRV guidelines.

Energy Requirement (Per Day)	2.290 kcal
Energy (kcal) mid-morning snack, 15% of 2.290 kcal	343.49 kcal
Fat	128.63 kcal
Saturated fats	36.75 kcal
Proteins	23.69 kcal
Carbohydrates	185.49 kcal
Sugars	34.35 kcal
Fibre	5.70 kcal
Salt (sodium chloride)	0.47 g

**Table 3 foods-12-00952-t003:** Formulations for the 4 types of chickpea biscuits studied.

Ingredient	G100:0	G75:25	G50:50	G25:75
Chickpea flour	53.3%	40.0%	26.7%	13.3%
Rice flour	0.0%	13.3%	26.7%	40.0%
Water	23.5%	23.5%	23.5%	23.5%
Sunflower oil	13.3%	13.3%	13.3%	13.3%
Orange blossom water	2.4%	2.4%	2.4%	2.4%
Sugar	7.2%	7.2%	7.2%	7.2%
Salt	0.3%	0.3%	0.3%	0.3%
Total	100.0%	100.0%	100.0%	100.0%

**Table 4 foods-12-00952-t004:** Nutritional composition of the four biscuits (mean value of 3 measurements and standard deviation are shown). G100:0 = 100% CF/0% RF; G75:25 = 75% CF/25% RF; G50:50 = 50% CF/50% RF; G25:75 = 25% CF/75% RF.

Nutrient	G100:0	G75:25	G50:50	G25:75
Energy (kJ/100 g)	1966.19 ± 5.16 ^a^	1945.40 ± 6.87 ^b^	1923.71 ± 6.80 ^c^	1928.94 ± 2.52 ^bc^
Energy (Kcal/100 g)	471.76 ± 5.16 ^a^	465.92 ± 6.87 ^a^	461.60 ± 6.80 ^a^	457.15 ± 2.52 ^a^
Protein (g/100 g)	13.92 ± 0.81 ^a^	10.54 ± 0.31 ^b^	9.89 ± 0.44 ^b^	7.71 ± 0.59 ^c^
Fat (g/100 g)	21.15 ± 2.39 ^a^	20.25 ± 0.88 ^a^	20.09 ± 0.33 ^a^	19.92 ± 1.26 ^a^
Saturated (g/100 g)	2.76 ± 1.01 ^a^	2.20 ± 0.08 ^a^	2.38 ± 0.21 ^a^	2.10 ± 0.21 ^a^
Monounsaturated (g/100 g)	8.60 ± 0.55 ^a^	9.05 ± 0.99 ^a^	10.59 ± 2.36 ^a^	8.13 ± 8.13 ^a^
Polyunsaturated (g/100 g)	8.39 ± 0.26 ^a^	8.88 ± 0.85 ^a^	8.32 ± 0.65 ^a^	8.05 ± 1.06 ^a^
Carbohydrates (g/100 g)	49.63 ± 4.19 ^b^	56.15 ± 2.70 ^ab^	56.56 ± 0.39 ^a^	62.27 ± 0.97 ^a^
Sugars (g/100 g)	18.84 ± 4.45 ^a^	13.64 ± 0.97 ^ab^	12.61 ± 2.34 ^ab^	11.60 ± 0.97 ^b^
Fibre (g/100 g)	7.07 ± 1.55 ^a^	4.82 ± 0.45 ^ab^	4.84 ± 0.93 ^ab^	2.59 ± 0.51 ^b^
Salt (g/100 g)	0.43 ± 0.077 ^a^	0.42 ± 0.071 ^a^	0.36 ± 0.094 ^a^	0.44 ± 0.071 ^a^
Sodium (mg/100 g)	166.89 ± 32.42 ^a^	176.03 ± 50.42 ^a^	154.38 ± 27.55 ^a^	151.08 ± 30.34 ^a^
Potassium (mg/100 g)	576.70 ± 20.10 ^a^	397.22 ± 2.22 ^b^	313.64 ± 5.11 ^c^	233.74 ± 7.52 ^d^
Calcium (mg/100 g)	29.00 ± 0.77 ^a^	24.03 ± 1.21 ^b^	18.99 ± 0.71 ^c^	14.23 ± 0.72 ^d^
Phosphorus (mg/100 g)	209.73 ± 8.93 ^a^	166.29 ± 4.61 ^c^	148.51 ± 2.28 ^b^	111.67 ± 11.10 ^b^
Magnesium (mg/100 g)	113.45 ± 2.35 ^a^	77.44 ± 9.50 ^b^	67.95 ± 8.40 ^bc^	54.12 ± 2.87 ^c^
Iron (mg/100 g)	3.39 ± 0.10 ^a^	2.37 ± 0.096 ^b^	1.77 ± 0.050 ^c^	1.26 ± 0.037 ^d^
Zinc (mg/100 g)	1.99 ± 0.059 ^a^	1.42 ± 0.056 ^b^	1.23 ± 0.023 ^c^	1.00 ± 0.027 ^d^
Copper (mg/100 g)	0.62 ± 0.079 ^a^	0.45 ± 0.008 ^b^	0.35 ± 0.020 ^c^	0.25 ± 0.020 ^d^
Manganese (mg/100 g)	1.10 ± 0.01 ^a^	1.01 ± 0.01 ^b^	0.97 ± 0.014 ^c^	0.93 ± 0.00 ^d^
Selenium (µg/100 g)	5.83 ± 0.09 ^c^	7.56 ± 0.27 ^b^	8.00 ± 0.25 ^b^	9.08 ± 0.02 ^a^

^abc^ Means with different letters in the same row are significantly different according to Tukey’s test at *p* < 0.05.

**Table 5 foods-12-00952-t005:** Mechanical and acoustic properties, baking loss, moisture content and water activity (aw) of the four chickpea biscuits (mean value of 3 measurements and standard deviation are shown). Parameters for mechanical properties: maximum force required to break (N), distance at break (mm), number of force peaks (peaks higher than 0.2 N) and linear distance (N*s). Parameters for acoustic properties: maximum sound pressure level (dB) and number of sound peaks. G100:0 = 100% CF/0% RF; G75:25 = 75% CF/25% RF; G50:50 = 50% CF/50% RF; G25:75 = 25% CF/75% RF.

Sample	G100:0	G75:25	G50:50	G25:75
Baking loss (%)	29.26 ^ab^ ± 0.88	29.21 ^ab^ ± 0.90	30.07 ^a^ ± 1.27	28.89 ^b^ ± 0.79
Moisture content (%)	3.96 ^b^ ± 0.30	3.59 ^b^ ± 0.25	3.95 ^b^ ± 0.28	6.50 ^a^ ± 0.40
aw	0.31 ^b^ ± 0.01	0.27 ^c^ ± 0.02	0.29 ^bc^ ± 0.01	0.47 ^a^ ± 0.03
Maximum force (N)	18.13 ^a^ ± 4.83	19.68 ^a^ ± 3.49	11.68 ^b^ ± 2.69	10.97 ^b^ ± 1.11
Distance at break (mm)	0.74 ^a^ ± 0.26	0.83 ^a^ ± 0.21	0.71 ^a^ ± 0.13	0.70 ^a^ ± 0.10
Number of force peaks	9.8 ^a^ ± 6.14	1.17 ^b^ ± 0.41	2.6 ^b^ ± 0.89	2.2 ^b^ ± 1.10
Linear distance (N*s)	94.76 ^a^ ± 13.28	81.35 ^b^ ± 7.19	68.48 ^c^ ± 4.72	76.40 ^bc^ ± 2.88
Maximum SPL (dB)	75.74 ^a^ ± 6.11	65.36 ^b^ ± 3.26	69.62 ^ab^ ± 6.79	67.99 ^b^ ± 5.34
Number of sound peaks	621.60 ^a^ ± 39.14	610.17 ^a^ ± 32.77	586.40 ^a^ ± 37.36	625.40 ^a^ ± 21.35

Means with different letters in the same row are significantly different according to Tukey’s test at *p* < 0.05.

**Table 6 foods-12-00952-t006:** Descriptors generated during the training of the sensory panel.

Attributes	Definition	Technique Used
Hardness	The force required to deform the product or to penetrate the product with a tool (e.g., a knife).	Place the sample between the incisors or between the tongue and the roof of your mouth.
Crispness	Sound with numerous acoustic events emitted by the product while chewing.	Place the sample between the incisors and evaluate the intensity of the sound during the first bite and while chewing (e.g., potato crisps).
Crunchiness	The sound produced with molars while chewing the product.	Place the product between the molars and evaluate the intensity of the sound emitted (e.g., nuts).
Grittiness	Geometric property related to the perception of the product particle size and shape.	Place the sample in the mouth and evaluate the thickness/size of the particles in the sample. The grittier (sand/dust-like particles), the higher the grade.
Fragility	Force required to break the sample into pieces.	Chew the sample and evaluate the force required to break it into pieces.
Prickliness/penetration	Perception of angular particles. They do not cause damage, but the edges are perceived.	Chew the sample and evaluate the geometrical shape of the particles. The sharper the particles, the higher the value on the grading scale.
Chewiness	A mechanical attribute of texture related to the effort required to chew a solid product until it is ready to be swallowed.	The number of bites needed to reduce the sample to a ready-to-swallow state. The more bites, the higher the grade value.

**Table 7 foods-12-00952-t007:** Adjusted means for each sample–attribute combination from the analysis of variance models. Letter “A” indicates values significantly higher than the global mean and letter “B” indicates values significantly smaller than the global mean. G100:0 = 100% CF/0% RF; G75:25 = 75% CF/25% RF; G50:50 = 50% CF/50% RF; G25:75 = 25% CF/75% RF.

	Prickliness	Crunchiness	Chewiness	Crispiness	Grittiness	Hardness	Fragility
G100:0	0.6	3.4 A	3.4 A	0.6	3.4 A	3.8 A	2.4 B
G75:25	0.6	2.6	3.2	1.2 A	3.2	3.4	2.6
G50:50	0.4	2.4 B	2.4 B	0.6	3.4 A	3.8 A	3.6 A
G25:75	0.6	3.2	2.6	0.6	2.4 B	3.0 B	2.8
Global mean	0.55	2.9	2.9	0.75	3.1	3.5	2.85

**Table 8 foods-12-00952-t008:** The average nutritional content of 49 different plain biscuits targeted at children (aged 5–12) sold in Spain between 2017 and 2022. Data found in Mintel GNPD Database (October 2022).

Nutrient (*n* = 49)	Content
Energy (kcal/100 g)	455.5
Energy (kJ/100 g)	1913.1
Carbohydrates (g/100 g)	70.3
Sugars (g/100 g)	21.6
Fat (g/100 g)	16.0
Fibre (g/100 g)	3.2
Protein (g/100 g)	6.0
Salt (g/100 g)	0.9

**Table 9 foods-12-00952-t009:** Micronutrient content per serving size (30 g of biscuit), % of AR covered by a serving size and DRVs for micronutrients recommended by EFSA, for adolescents from 11 to 14 years, engaged in all types of physical activity. The DRVs indicated are the UL (Tolerable Upper Intake Level), which is “the maximum level of total chronic intake of a nutrient from all sources judged to be unlikely to pose a risk of adverse health effects in humans”; and the AR (Average Requirement), which refers to “the intake of a nutrient that meets the daily needs of half the people in a typical healthy population”. G100:0 = 100% CF/0% RF; G75:25 = 75% CF/25% RF; G50:50 = 50% CF/50% RF; G25:75 = 25% CF/75% RF.

	mg per Serving Size (30 g Biscuit)				% AR Covered by 30 g of Biscuit				AR (mg/day)	UL (mg/day)
	G100:0	G75:25	G50:50	G25:75	G100:0	G75:25	G50:50	G25:75		
Iron	1.02	0.71	0.531	0.378	12.71%	8.89%	6.64%	4.73%	8	-
Potassium	173.01	119.17	94.092	70.122	6.41%	4.41%	3.48%	2.60%	2700 *	-
Zinc	0.60	0.43	0.369	0.300	6.71%	4.79%	4.15%	3.37%	8.9	18
Phosphorus	62.92	49.89	44.553	33.501	9.83%	7.79%	6.96%	5.23%	640 *	-
Magnesium	34.04	23.23	20.385	16.236	13.61%	9.29%	8.15%	6.49%	250 *	250
Copper	0.19	0.14	0.105	0.075	16.91%	12.27%	9.55%	6.82%	1.1 *	4
Manganese	0.33	0.30	0.291	0.279	16.50%	15.15%	14.55%	13.95%	2 *	-
Calcium	8.70	7.21	5.697	4.269	0.91%	0.75%	0.59%	0.44%	960	-
Selenium **	1.75	2.27	2.40	2.72	3.18%	4.12%	4.36%	4.95%	55 *	200

* DRV indicated as AI (adequate intake), which is used when there is not enough data to calculate an average requirement. An AI is “the average nutrient level, based on observations or experiments, that is assumed to be adequate for the population’s needs”. ** expressed in µg.

## Data Availability

The data that support the findings of this study are available as Appendix A upon request.

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
