# Peer review of "Rational Food Design Targeting Micronutrient Deficiencies in Adolescents: Nutritional, Acoustic-Mechanical and Sensory Properties of Chickpea-Rice Biscuits"

_foods, 2023, doi:10.3390/foods12050952_

Round 1
Reviewer 1 Report
The paper “Rational Food Design Targeting Micronutrient Deficiencies in Adolescents: Nutritional, Acoustic-Mechanical and Sensory Properties of Chickpea-Rice Biscuits” is focused on assessing the nutritional and sensory characteristics of biscuits obtained from blends of chickpea and rice flours. Also, the perception of students from secondary school (adolescents) on these biscuits is tested in order to find out the suitability of such biscuits as a mid-morning snack.
The paper contains relevant information. The qualitative parameters of chickpea-rice biscuits are well chosen and determined using adequate analytical methods.
However, as the paper title and aim indicate, the Rational Food Design (RDF) is the approach that the authors have been chosen to achieve their research goal and the article did not present how RDF approach was used. This is the main weakness of the article.
The introduction section contains relevant information, but it is necessary to be completed with more information about RDF approach.
The materials and methods section must be completed with the analytical method used for salt, water activity and baking loss determination. These parameters are discussed in the Results section but without presenting their methodological determination.
The Results and discussion section need to be written by indicating how the Rational Food Design approach was used to achieve the research goal. Otherwise, the results are presented only in a general manner and it is quite predictable that by using a chickpea-rise blend in biscuits manufacturing their nutritional quality will be improved, comparing to wheat flour biscuits.
Regarding the sentence from lines 385-386 -The "crispiness" and "prickliness" attributes could not be used to distinguish the samples from each other-please indicate if this conclusion was obtained from Figure 3 or Table 8. It is not clear.
For the sensorial evaluation, the authors must indicate which is the optimal sample. I suppose that is the G 50:50 sample which was used further in consumer testing, but it needs to be indicated clearly.
The Discussion section, in particular, need to be rewritten in order to show how RDF is useful for micronutrient deficiencies in the case of these biscuits.
Author Response
The authors would like to acknowledge the reviewer for his/her valuable comments, that have increased the quality of the manuscript. Please find below the replies to the comments.
The paper “Rational Food Design Targeting Micronutrient Deficiencies in Adolescents: Nutritional, Acoustic-Mechanical and Sensory Properties of Chickpea-Rice Biscuits” is focused on assessing the nutritional and sensory characteristics of biscuits obtained from blends of chickpea and rice flours. Also, the perception of students from secondary school (adolescents) on these biscuits is tested in order to find out the suitability of such biscuits as a mid-morning snack.
The paper contains relevant information. The qualitative parameters of chickpea-rice biscuits are well chosen and determined using adequate analytical methods.
However, as the paper title and aim indicate, the Rational Food Design (RFD) is the approach that the authors have been chosen to achieve their research goal and the article did not present how RFD approach was used. This is the main weakness of the article.
The introduction section contains relevant information, but it is necessary to be completed with more information about RFD approach.
A paragraph explaining RFD approach is now provided under Materials and Methods section, to avoid extending the length of the introduction section. It has also been explained in the Results and Discussion sections.
The materials and methods section must be completed with the analytical method used for salt, water activity and baking loss determination. These parameters are discussed in the Results section but without presenting their methodological determination.
A new section has been renamed as “physico-chemical analysis”, including salt, water activity and baking loss determination.
The Results and discussion section need to be written by indicating how the Rational Food Design approach was used to achieve the research goal. Otherwise, the results are presented only in a general manner and it is quite predictable that by using a chickpea-rise blend in biscuits manufacturing their nutritional quality will be improved, comparing to wheat flour biscuits.
A paragraph has been included at the start of the Results section about how the Rational Food Design approach was used.
Regarding the sentence from lines 385-386 -The "crispiness" and "prickliness" attributes could not be used to distinguish the samples from each other-please indicate if this conclusion was obtained from Figure 3 or Table 8. It is not clear.
We have rephrased the sentence and indicated that the conclusion comes from Figure 3:
For the sensorial evaluation, the authors must indicate which is the optimal sample. I suppose that is the G 50:50 sample which was used further in consumer testing, but it needs to be indicated clearly.
The sensory analysis was carried out with a trained panel with the objective of carrying out a qualitative descriptive analysis (QDA). As it was an expert panel, questions regarding preference cannot be included.
The Discussion section, in particular, need to be rewritten in order to show how RDF is useful for micronutrient deficiencies in the case of these biscuits.
A paragraph has been included at the start of the Discussion section about how the Rational Food Design approach was used.

Reviewer 2 Report
Line 48: Please elaborate ‘AVENA’
Line 138-39: Please elaborate ‘RDF’
Introduction is too long, which may be briefed.
Was any other additive (such as raising agent, flavor, color) used in the recipe? If no raising agent was used then how the texture of biscuits was maintained, particular biscuit texture?
Some recent and relevant citations may be added such as:
· Quality attributes of Cookies Prepared with Moringa (Moringa oleifera) Seed Oil (https://pjlss.edu.pk/pdf_files/2021_1/7-14.pdf)
Author Response
The authors would like to acknowledge the reviewer for his/her valuable comments, that have increased the quality of the manuscript. Please find below the replies to the comments.
Line 48: Please elaborate ‘AVENA’
AVENA acronym has been explained
Line 138-39: Please elaborate ‘RDF’
RFD acronym has been corrected.
Introduction is too long, which may be briefed.
Introduction length has been reduced.
Was any other additive (such as raising agent, flavor, color) used in the recipe? If no raising agent was used then how the texture of biscuits was maintained, particular biscuit texture?
There was no raising agent or other additives added to the formula. The authors considered that there was no need of addition for the purpose of the study.
Some recent and relevant citations may be added such as:
- Quality attributes of Cookies Prepared with Moringa (Moringa oleifera) Seed Oil (https://pjlss.edu.pk/pdf_files/2021_1/7-14.pdf)
We appreciate the suggestion, but we think this citation cannot be related to a result of our work.

Reviewer 3 Report
The article is presented in a very well structured and elaborate manner. However, there is scope for improvement.
1. Introduction can be concise, relevant and crisp.
2.Materials and Methods- Is the CF used raw or is any pretreatment given before preparing the actual product?
3.What is the shelf life of the product?
4. Was the Ethics committee approval obtained before consumer testing?
5. Consumer tasting - noticeably young population for sensory evaluation was selected. This population is prone to peer pressure and the answers given might not be reliable. What are the precautions taken to overcome this limitation?
- Sample size of 33 children is too low to conclude the suitability and generalizability of the product. - as only 36 % were able to provide any response for flavor.
- Was any allergic reaction / abdominal discomfort reported during testing?
Results:
6. Table 5- Anti nutritional compounds such as phytate levels can be reported.
Discussion:
7.Anti nutritional compounds - such as enzymes / non enzyme inhibitors to be described in discussion section as it will be a major issue in targeting micronutrient deficiencies.
8.Limitations and strengths of the study to be described.
9.Baking losses of micronutrients also to be incorporated in discussion section with relevant references.
Conclusion:
10. Precise conclusions to be added as per the aim of the study.
Author Response
The authors would like to acknowledge the reviewer for his/her valuable comments, that have increased the quality of the manuscript. Please find below the replies to the comments:
The article is presented in a very well structured and elaborate manner. However, there is scope for improvement.
- Introduction can be concise, relevant and crisp.
Introduction length has been reduced.
2.Materials and Methods- Is the CF used raw or is any pretreatment given before preparing the actual product?
The CF (chickpea flour) used was obtained from 100% chickpeas clean and milled (as per technical specification from the supplier). If there was a pre-treatment we would have indicated it in the article.
3.What is the shelf life of the product?
The main aim of the study was not analysing chickpea’s shelf-life, but we can consider this as future work. Thanks for the suggestion.
- Was the Ethics committee approval obtained before consumer testing?
Yes, the ethical approval was conducted according to quality standards of the ISO 8586:2012 certified by AENOR, the Spanish Association for Standardization and Certification.
- Consumer tasting - noticeably young population for sensory evaluation was selected. This population is prone to peer pressure and the answers given might not be reliable. What are the precautions taken to overcome this limitation?
The participants assessed the samples in inviduals cabins without influence of peers. We have included the following sentence: The participants assessed the samples in individual cabins illuminated by fluores-cent lamps. The sample was served in individual plates
- Sample size of 33 children is too low to conclude the suitability and generalizability of the product. - as only 36 % were able to provide any response for flavor.
Other published studies have also used a similar sample size in studies with adolescents:
Høgsdal, H., Kaiser, S., & Kyrrestad, H. (2023). Adolescents’ assessment of two mental health–promoting mobile apps: results of two user surveys. JMIR Formative Research, 7(1), e40773.
- Was any allergic reaction / abdominal discomfort reported during testing?
No allergic reaction/ abdominal discomfort was reported
Results:
- Table 5- Anti nutritional compounds such as phytate levels can be reported.
The analysis of antinutritional compounds was not carry out as we used a commercial chickpea flour (we did not prepare it ourselves). However, we can consider this as future work. Thanks for the suggestion.
Discussion:
7.Anti nutritional compounds - such as enzymes / non enzyme inhibitors to be described in discussion section as it will be a major issue in targeting micronutrient deficiencies.
The objective of the study was to use commercial ingredients fit for human consumption. We acknowledge that the presence of antinutritional factors is relevant to target micronutrient deficiencies, but it was out of scope of the main aim of the study, which was applying the rational food design concept. However, we will consider this aspect in future works.
8.Limitations and strengths of the study to be described.
Limitations and strengths are included in the discussion.
9.Baking losses of micronutrients also to be incorporated in discussion section with relevant references.
The minerals were analysed after baking. We did not measure the micronutrient content of the dough, so we cannot discuss if there are micronutrient losses during baking.
Conclusion:
- Precise conclusions to be added as per the aim of the study.
Conclusions have been re-written more precisely.

Reviewer 4 Report
Dear Sir / Madam,
Please find below my comments and suggestions following the review of the study titled ‘Rational Food Design Targeting Micronutrient Deficiencies in 2 Adolescents: Nutritional, Acoustic-Mechanical and Sensory 3 Properties of Chickpea-Rice Biscuits’.
The authors have chosen an interesting topic to study. The topic chosen i.e. developing a nutritious snack for adolescents is a relevant one. I commend the authors on the work they have undertaken. This study has been logically planned and systematically executed. Also, the manuscript is very well-written. The following are some minor revisions that I suggest:
1. Introduction:
· Please modify the sentence “Importantly, chickpeas have high levels of micronutrients usually lacking in adolescence.” as follows “Importantly, chickpeas have high levels of micronutrients usually lacking in diets consumed by adolescents.”
· Line 138 – The acronym should read RFD and not RDF.
2. Materials and Methods:
· Line 154 – Biscuits were formulated using different ratios and not at different ratios
· Line 225 – What do the authors mean by “complexity of tasting sessions with children”? Please elaborate.
· What does the letter ‘G’ before the flour ratios of different biscuit stand for?
3. Results:
· In Table 5, 6 and 8, please ensure that all the letters indicating differences are placed in superscript.
· Line 270 - In the sentence “a tendency to increase the sugar and fibre content was observed for biscuits with larger amounts of CF.” when the authors say a tendency to increase sugar and fibre content was observed, do you mean that additional sugar was added while preparing the biscuits. But you also mention that amounts of all other ingredients in the preparation of the biscuits was fixed and only the amounts of the flours were different. This sentence then is confusing to the reader. Please modify it. You may want to consider calling the sugar naturally present in chickpea flour ‘natural sugars’ and that added to the biscuit dough, ‘added sugar’, to make the meaning clear to the readers.
· Similarly in Line 294 “There is a tendency to increase the energy content as the proportion of added CF increases.” The authors may consider modifying this sentence to read as follows - There is a tendency for the energy content to increase as the proportion of added CF increases
· Line 283 – Please specify that it was possible to reach 100% of the DRV for nutritional requirements in mid-morning snack.
· What was the serving size of the biscuit? How much of the DRV was reached in one serving size for energy, macro- and micronutrients?
4. Discussion:
· It is important to discuss about the %DRV that can be reached when one serve of biscuit is consumed. It seems that while the nutritional composition of 100g of the biscuits reached the DRV, if one serve of biscuits is considered to be 30g, a serve may fall considerably short of the DRV and then the adolescent would need to consume the biscuits in combination with another snack, like milk. In this case, the authors may also consider mentioning how such snack combinations could help meet the mid-morning snack DRV.
· Another point to consider for improving the acceptability of the biscuits is the use of cocoa as a flavouring but this could necessitate increasing the proportion of added sugar and fat. Addition of nuts may also be considered to improve the texture and flavour of the biscuits.
· The conclusion should also state how the flavour and texture affected the overall liking of the biscuits.
· Please also state limitations and strengths of your study and future scope for improvements.

Author Response
The authors would like to acknowledge the reviewer for his/her valuable comments, that have increased the quality of the manuscript. Please find below the replies to the comments.
Dear Sir / Madam,
Please find below my comments and suggestions following the review of the study titled ‘Rational Food Design Targeting Micronutrient Deficiencies in 2 Adolescents: Nutritional, Acoustic-Mechanical and Sensory 3 Properties of Chickpea-Rice Biscuits’.
The authors have chosen an interesting topic to study. The topic chosen i.e. developing a nutritious snack for adolescents is a relevant one. I commend the authors on the work they have undertaken. This study has been logically planned and systematically executed. Also, the manuscript is very well-written. The following are some minor revisions that I suggest:
- Introduction:
- Please modify the sentence “Importantly, chickpeas have high levels of micronutrients usually lacking in adolescence.” as follows “Importantly, chickpeas have high levels of micronutrients usually lacking in diets consumed by adolescents.”
corrected
- Line 138 – The acronym should read RFD and not RDF.
corrected
- Materials and Methods:
- Line 154 – Biscuits were formulated using different ratios and not at different ratios
corrected
- Line 225 – What do the authors mean by “complexity of tasting sessions with children”? Please elaborate.
This sentence has been elaborated.
- What does the letter ‘G’ before the flour ratios of different biscuit stand for?
G stands for Galleta in Spanish, so we kept the G. It is just sample coding.
- Results:
- In Table 5, 6 and 8, please ensure that all the letters indicating differences are placed in superscript.
Superscripts have been added in Table 5, 6. The legend of table 8 has been changed accordingly.
- Line 270 - In the sentence “a tendency to increase the sugar and fibre content was observed for biscuits with larger amounts of CF.” when the authors say a tendency to increase sugar and fibre content was observed, do you mean that additional sugar was added while preparing the biscuits. But you also mention that amounts of all other ingredients in the preparation of the biscuits was fixed and only the amounts of the flours were different. This sentence then is confusing to the reader. Please modify it. You may want to consider calling the sugar naturally present in chickpea flour ‘natural sugars’ and that added to the biscuit dough, ‘added sugar’, to make the meaning clear to the readers.
The sentence has been rephrased.
- Similarly in Line 294 “There is a tendency to increase the energy content as the proportion of added CF increases.” The authors may consider modifying this sentence to read as follows - There is a tendency for the energy content to increase as the proportion of added CF increases
The sentence has been rephrased.
- Line 283 – Please specify that it was possible to reach 100% of the DRV for nutritional requirements in mid-morning snack.
The DRV in-mid morning snacks are recommended per calories intake, not per micronutrient content. However, we have provided the baseline for our estimates.
- What was the serving size of the biscuit? How much of the DRV was reached in one serving size for energy, macro- and micronutrients?
An explanation gas been provided for the discussion on how the DRV for the different minerals is met.
- Discussion:
- It is important to discuss about the %DRV that can be reached when one serve of biscuit is consumed. It seems that while the nutritional composition of 100g of the biscuits reached the DRV, if one serve of biscuits is considered to be 30g, a serve may fall considerably short of the DRV and then the adolescent would need to consume the biscuits in combination with another snack, like milk. In this case, the authors may also consider mentioning how such snack combinations could help meet the mid-morning snack DRV.
We highly agree with this comment and acknowledge that the discussion was a bit confusing. A new table has been provided per serving size, per biscuit and per micronutrient.
- Another point to consider for improving the acceptability of the biscuits is the use of cocoa as a flavouring but this could necessitate increasing the proportion of added sugar and fat. Addition of nuts may also be considered to improve the texture and flavour of the biscuits.
Thanks for the suggestion, we can include this as limitations and future work.
- The conclusion should also state how the flavour and texture affected the overall liking of the biscuits.
Statement added in conclusions.
- Please also state limitations and strengths of your study and future scope for improvements.
Limitations and strengths are included in the discussion.
Reviewer 5 Report
The concept of the manuscript is good, experiments are well designed, results are presented clearly. However, a few minor changes would improve the quality of paper.
Line 26: Abstract: the maximum score of sensory analysis should be indicated
Introduction is interesting, but too long. Line 57-86 could be skipped
Table 3.: The range of data would be also interesting.
Line 119-133: description of nutritional composition af chickpea does not contain the numerical data. Please complete it.
Table 5.: recommended or maximum daily intake of minerals should be indicated. For selenium 5-9 milligrams per 100 grams seems to be very high.
Lines 283-287: Please indicate the amount of bisquit that equal to the DRV recommended by EFSA. Is it a piece of biscuit or 100 g?
Sensory profile of bisquits: seven panelsits seems to be too few to describe the sensory quality of bisquits. In Table 8. global mean should be indicated.
Consumer tasting: why was just one sample chosen to analyse? Is it sure that the selected sample was the best choice for this analyses?
Author Response
The authors would like to acknowledge the reviewer for his/her valuable comments, that have increased the quality of the manuscript. Please find below the replies to the comments.
The concept of the manuscript is good, experiments are well designed, results are presented clearly. However, a few minor changes would improve the quality of paper.
Line 26: Abstract: the maximum score of sensory analysis should be indicated
The maximum score of sensory analysis has been indicated
Introduction is interesting, but too long. Line 57-86 could be skipped
Introduction length has been reduced.
Table 3.: The range of data would be also interesting.
The range of data is indicated in the table caption.
Line 119-133: description of nutritional composition af chickpea does not contain the numerical data. Please complete it.
Numerical data has been included
Table 5.: recommended or maximum daily intake of minerals should be indicated. For selenium 5-9 milligrams per 100 grams seems to be very high.
A new table has been provided with the required information
Lines 283-287: Please indicate the amount of bisquit that equal to the DRV recommended by EFSA. Is it a piece of biscuit or 100 g?
A new discussion section has been re-written and calculations made based on a portion size of 30g.
Sensory profile of bisquits: seven panelsits seems to be too few to describe the sensory quality of bisquits. In Table 8. global mean should be indicated.
We have published other studies with 7-8 trained panellists. For a trained panel is statistically significant:
Talens C, Lago M, Illanes E et al. Development of the lexicon, trained panel validation and sensory profiling of new ready-to-eat plant-based "meatballs" in tomato sauce. Open Res Europe 2022, 2:145 (https://doi.org/10.12688/openreseurope.15360.1).
Hybrid Sausages: Modelling the Effect of Partial Meat Replacement with Broccoli, Upcycled Brewer’s Spent Grain and Insect Flours
C Talens, R Llorente, L Simó-Boyle, I Odriozola-Serrano, I Tueros, ...
Foods 11 (21), 3396, https://www.mdpi.com/2304-8158/11/21/3396
Desirability-based optimization of bakery products containing pea, hemp and insect flours using mixture design methodology
C Talens, M Lago, L Simó-Boyle, I Odriozola-Serrano, M Ibargüen
LWT 168, 113878, https://www.sciencedirect.com/science/article/pii/S0023643822008131
Global mean has been included in Table 8.
Consumer tasting: why was just one sample chosen to analyse? Is it sure that the selected sample was the best choice for this analyses?
The selection of the sample was based in the results of the texture analyses.
We have explained it in the following paragraph:
“This sample had the best texture for children aged between 10 and 12. It was the easiest to chew (the least effort required to chew the biscuit before swallowing) and the most fragile (the least force needed to break it into pieces) (see Tables 6 and 8)”.

Round 2
Reviewer 1 Report
The manuscript "Rational Food Design Targeting Micronutrient Deficiencies in Adolescents: Nutritional, Acoustic-Mechanical and Sensory Properties of Chickpea-Rice Biscuits" was improved according to the comments but still has some issues regarding the applied methodology and the Results and Discussion.
Part of the information of the Introduction section was moved to the Materials and Methods section aiming to describe RFD methodology. Hoewever, in my opinion Table 3 and Figure 1 are not suitable for this section.
The authors should explain the methodology of RFD better. Moreover, it is no clear how RFD could be apply to create acoustic properties in biscuits, as it was stated in lines 498-500
Line 540- please explain how you got the 30g size of serving biscuits?
The discussion of the results must be developed in relation to the RFD approach. It was not improved enough.
Author Response
We acknowledge the effort of the reviewer in highlighting the relevance of RFD. We have explained how we understand RFD, and how with this study we aim to widen the use of RFD.
Please find below the reply to the reviewer's comments:
REVIEW 1
The manuscript "Rational Food Design Targeting Micronutrient Deficiencies in Adolescents: Nutritional, Acoustic-Mechanical and Sensory Properties of Chickpea-Rice Biscuits" was improved according to the comments but still has some issues regarding the applied methodology and the Results and Discussion.
Part of the information of the Introduction section was moved to the Materials and Methods section aiming to describe RFD methodology. Hoewever, in my opinion Table 3 and Figure 1 are not suitable for this section.
Table 3 and Figure 1 have been moved to the discussion section, as seen in other similar publications where the Mintel database is used for discussion.
Elizabeth Cole, Natalie Goeler-Slough, Allison Cox & Alissa Nolden (2022) Examination of the nutritional composition of alternative beef burgers available in the United States, International Journal of Food Sciences and Nutrition, 73:4, 425-432, DOI: 10.1080/09637486.2021.2010035
The authors should explain the methodology of RFD better. Moreover, it is no clear how RFD could be apply to create acoustic properties in biscuits, as it was stated in lines 498-500
We have added some concepts to the definition of RFD:
(…)
In this study, we suggest to widen the use of RFD for targeted nutrition (aimed at the nutritional needs of specific groups). For that, RFD could be applied in two steps of the food development process:
- during the formula design step, which is also part of the process of structure design, when two important decisions are made: (1) the target nutritional composition of the new or improved formula, as well as (2) the ingredients and raw materials used based on their macro- and micronutrients composition.
- during the step preceding the consumer tasting, when a selection of the optimum samples has to be made based on acoustic-textural and sensory analysis.
(…)
From what we have seen in the literature there is not an official definition for the Rational Food Design. Every research group decides which disciplines brings to the RFD. In our approach, we decided to bring the scientific knowledge provided by EFSA’s DRV, as well as the acoustic-texture measurements as “rationals” to compare and select among different food formula designs.
Line 540- please explain how you got the 30g size of serving biscuits?
We have explained in the text, it was based on the average size of the commercial samples found in the Mintel search.
The discussion of the results must be developed in relation to the RFD approach. It was not improved enough.
Same as above. We have stated in the results how we achieve the new 2 concepts brought to widen the use of RFD in food product design and development.

Reviewer 3 Report
1. All the macronutrients can be rounded off in the respective table.
2. Limitations and strength should be before conclusion section.
Author Response
The authors would like to acknowledge again the reviewer during the second round of the revision process.
In respond to the comments:
1. All the macronutrients can be rounded off in the respective table
Table 6 has been updated with 2 decimals for the micronutrient contents.
2. Limitations and strength should be before conclusion section.
We appreciate the suggestion, however, we believe "limitations and strength" should be part of the conclusion section, as it states future work, and leaves the door open to continue or complement our research.